# Barriers facing persons with disability in accessing sexual and reproductive health services in sub-Saharan Africa: A systematic review

John Kuumuori Ganle[1,2]*, Leonard Baatiema[3], Reginald Quansah[4], Anthony Danso-Appiah[5]

1 Department of Population, Family and Reproductive Health, School of Public Health, University of Ghana, Accra, Ghana, 2 Stellenbosch Institute for Advanced Study, Stellenbosch, South Africa, 3 Noguchi Memorial Institute for Medical Research, University of Ghana, Accra, Ghana, 4 Department of Biological, Environmental and Occupational Health, School of Public Health, University of Ghana, Accra, Ghana, 5 Department of Epidemiology and Disease Control, School of Public Health, University of Ghana, Accra, Ghana

* jganle@ug.edu.gh

**Data Availability Statement:** All relevant data are within the paper.

**Funding:** The author(s) received no specific funding for this work.

## Abstract

### Background

There is evidence that persons with disabilities often encounter grave barriers when accessing sexual and reproductive health services. To the best of our knowledge, however, no systematic review has been conducted to pull together these pieces of research evidence for us to understand the nature, magnitude and extent of these barriers in different settings in sub-Saharan Africa. We do not yet have a good understanding of the strength/quality of the evidence that exist on the barriers persons with disabilities face when accessing sexual and reproductive health services in sub-Saharan Africa. We therefore conducted a systematic review to examine the barriers persons with disabilities face in accessing sexual and reproductive health services in sub-Saharan Africa.

### Methods

A systematic review was conducted using PRISMA guidelines (PROSPEROO protocol registration number: CRD42017074843). An electronic search was conducted in Medline, EMBASE, CINAHL, PsycINFO, and Web of Science from 2001 to 2020. Manual search of reference list was also conducted. Studies were included if they reported on barriers persons with disability face in accessing sexual and reproductive health services. The Critical Appraisal Skills Programme and Centre for Evidence Based Management (CEBMa) appraisal tools were used to assess methodological quality of eligible studies.

### Findings

A total of 1061 studies were identified. Only 26 studies covering 12 sub-Saharan African countries were eligible for analysis. A total of 33 specific barriers including inaccessible

**Competing interests:** The authors have declared that no competing interests exist.

physical health infrastructure and stigma and discrimination were identified. These barriers were further categorised into five levels: broader national level barriers; healthcare system/ institutional barriers; individual level barriers; community level barriers; and economic barriers.

## Conclusion

Persons with disabilities face a myriad of demand and supply side barriers to accessing sexual and reproductive healthcare in sub-Saharan Africa. Multilevel interventions are urgently needed to address these barriers.

## Background

Persons with disabilities (PWDs) constitute more than 15% of the world's population [1]. Disability is the consequence of an impairment that could be physical, cognitive, mental, sensory, emotional, developmental, or some combination of these that result in restrictions on an individual's ability to participate in their everyday society [1].

PWDs are one of the most marginalised and socially excluded groups in many countries [2–4]. This marginalization transcends several spheres: PWDs have generally poorer health, lower education achievements, fewer economic opportunities and higher rates of poverty than people without disabilities [1]. In particular, women with disability are more likely to be poorer and have lower social and economic status than their counterparts who have no disability [3–5]. In recognition of this, the United Nations Convention on the Rights of Persons with Disabilities guarantees PWDs the fundamental human rights and equitable opportunities to access quality and standard of healthcare [2]. In spite of increased awareness created by the UN Convention, PWDs still face numerous challenges to accessing healthcare [4–11]. Impediments to accessing healthcare services include attitudinal biases of health and social service providers, and physical barriers in clinical settings [5–7, 9, 12, 13].

In the context of sexual and reproductive health, a number of recent studies note that PWDs have been ignored in many low-income settings [14–23]. Part of the reason for this neglect is the impression that PWDs are not sexually active and less likely to marry or have children than persons without disability [14–23]. Recent evidence however shows that rates of sexual desire and activity, need for family planning services, and childbearing among disabled women are comparable to those of non-disabled women [1, 19, 23]. In this regard, it is noteworthy that a number of studies within sub-Saharan Africa have started to highlight the challenges PWDs face accessing sexual and reproductive health information and services [7, 9, 11, 18–20, 22]. While these studies provide useful evidence on the barriers to accessing sexual and reproductive health services among PWDs in the individual contexts within which they have been conducted, no systematic review has been conducted to pull together these pieces of research evidence for us to understand the nature, magnitude and extent of these barriers across sub-Saharan Africa. Moreover, we do not yet have a good understanding of the strength/quality of the evidence that exist on the barriers persons with disabilities face when accessing sexual and reproductive health services in sub-Saharan Africa. This evidence gap could potentially undermine sub-regional planning and efforts to develop more inclusive sexual and reproductive healthcare policies and programmes that have the potential to propel progress towards the Sustainable Development Goals' 3 objective of universal and/ or equitable access to skilled and comprehensive sexual, reproductive and maternal health services. To address gap, this systematic review aimed to answer the following inter-related research

questions: what is the evidence that PWDs face barriers in accessing sexual and reproductive health (SRH) information and services in sub-Saharan Africa; and what specific barriers do PWDs face in accessing sexual and reproductive health (SRH) information and services in sub-Saharan Africa?

## Materials and methods

The review was conducted according to the standards and good practices of preparing a systematic review [24–26]. The conduct and reporting of the review was done in accordance with the PRISMA guideline for reporting systematic reviews and meta-analysis [27]. The protocol for the systematic review was registered in PROSPEROO (registration number: CRD42017074843).

### Criteria for considering studies for this review

**Types of studies.**   Both quantitative and qualitative studies published between 2001 and 2020 were eligible for inclusion in the review. Specifically, studies using such data collection techniques as in-depth interviews, focus groups discussions and surveys that have been conducted at a primary healthcare setting, hospital or community level in sub-Saharan Africa and assessed barriers PWDs face in accessing sexual and reproductive health services were included. Only peer reviewed journal articles were considered. Commentaries, editorials, letters written to editors or policy statements were excluded. The year 2001 was chosen to correspond with the period the UN General Assembly established an Ad Hoc Committee to consider proposals for a comprehensive convention to promote and protect the rights and dignity of persons with disabilities. The work of this Ad Hoc Committee culminated in the adoption of the UN Convention on the Rights of Persons with Disabilities in December 2006, which increased global attention to issues affecting PWDs.

**Types of population.**   Persons with disability in this study included those with physical and sensory impairments, developmental and intellectual disability and psychosocial disability. For inclusion, studies must have been conducted in any country in sub-Saharan Africa, and involve either male or female PWDs who are aged 15years and above. While both age of menarche among girls and sexual debut among boys and girls have declined in recent years, most international policy and research on sexual and reproductive health often focus on age 15 onwards as the starting point of sexual activity and reproduction [11, 17, 18]. Our focus on 15years and above was therefore informed by this international policy and research literature. Studies which reported the views of healthcare personnel who provide direct sexual and reproductive healthcare services to PWDs as well as community and family members of PWDs were also eligible for inclusion.

**Types of intervention.**   Studies which sought to identify barriers PWDs face in accessing sexual and reproductive health services were included. Specifically, PWDs should have accessed or likely to access one of the following: sexual health education and information, family planning, contraception, abortion, antennal care (ANC), health facility childbirth, and postnatal care (PNC) services.

**Outcomes of interest.**   The outcomes of interest in this review included perceived and actual barriers or challenges PWDs face in accessing sexual and reproductive health services. Such barriers should relate to access to or use of sexual health education and information, family planning, contraception, abortion, ANC, health facility childbirth, and PNC services.

### Search strategy

We searched five electronic databases, namely MEDLINE, EMBASE, CINAHL, PsycINFO and Web of Science from 2001 to March 2020 with only English Language restriction. The

choice of these databases was based on their indexing coverage of biomedical and allied health journals related to the review topic. A Medline search strategy was developed and subsequently adapted and applied to the other databases using the appropriate MeSH or key terms. The reference lists of retrieved studies were screened for additional potentially relevant studies. The search strategy and search terms are reported in **S1 File**.

### Study selection

Two authors searched the results from the five debases. Articles were exported to Endnote reference manager where duplicates were removed. The selection process was systematically conducted and displayed in flow chart in line with the PRISMA guidelines (**see Fig 1**). First, the titles and abstracts of studies were screened using pretested study selection form developed from the inclusion criteria to identify potentially eligible studies. Second, all potentially relevant studies' titles and abstracts were identified by two authors. The remaining authors independently sampled at least five of the eligible studies following title and abstract screening. To minimize bias, authors did not review prospective studies where they were authors. Full text screening was further conducted by two authors and where disagreement arose about the potential eligibility of a particular article, a third reviewer was involved. Where eligible studies reported insufficient information to support the review process, the corresponding authors of those articles were contacted by one reviewer. All studies which did not meet the eligibility criteria were excluded with the reasons for exclusion provided. Where some eligible studies had missing data or presented insufficient published data, one reviewer contacted the study's corresponding authors to clarify the missing data and retrieve same where the data was able.

### Data extraction

In order to ensure consistency and transparency, the data extraction process was facilitated by a standardized evidence table (**see Table 1**) where data on the study's author (s), setting, aim, study design, methods, population characteristics and key findings were extracted. This was done by two reviewers and where a discrepancy arose, the other two reviewers were invited to resolve the issue before the extraction process proceeded.

### Quality assessment

Quality appraisal of eligible studies assessed the study design, study aim, sampling procedures, role of confounding factors for potential bias and potential generalisability of findings using two widely used best practice quality appraisal tools: the CASP checklist [28], and the quality assessment tool for surveys by the Center for Evidence-based Management (CEBMa) [29]. For all mixed-methods studies, the applicable quality appraisal tool was employed depending on the study design. These tools (CEBMa and CASP) are well established, scientifically rigorous and widely used and thus their external validity is not in doubt. The first and second authors of this paper led the quality assessment. Where discrepancies arose, a third reviewer acted as an arbiter. To optimize objectivity in the quality appraisal process, where reviewers are authors of eligible studies, they were not included in the quality assessment process.

### Data synthesis

A qualitative synthesis approach was used. Findings were presented narratively and in tables. To enhance reporting transparency, the framework for data synthesis by Popay et al [30] was used. Here, data were reported using tables, highlighting key and unique barriers to accessing

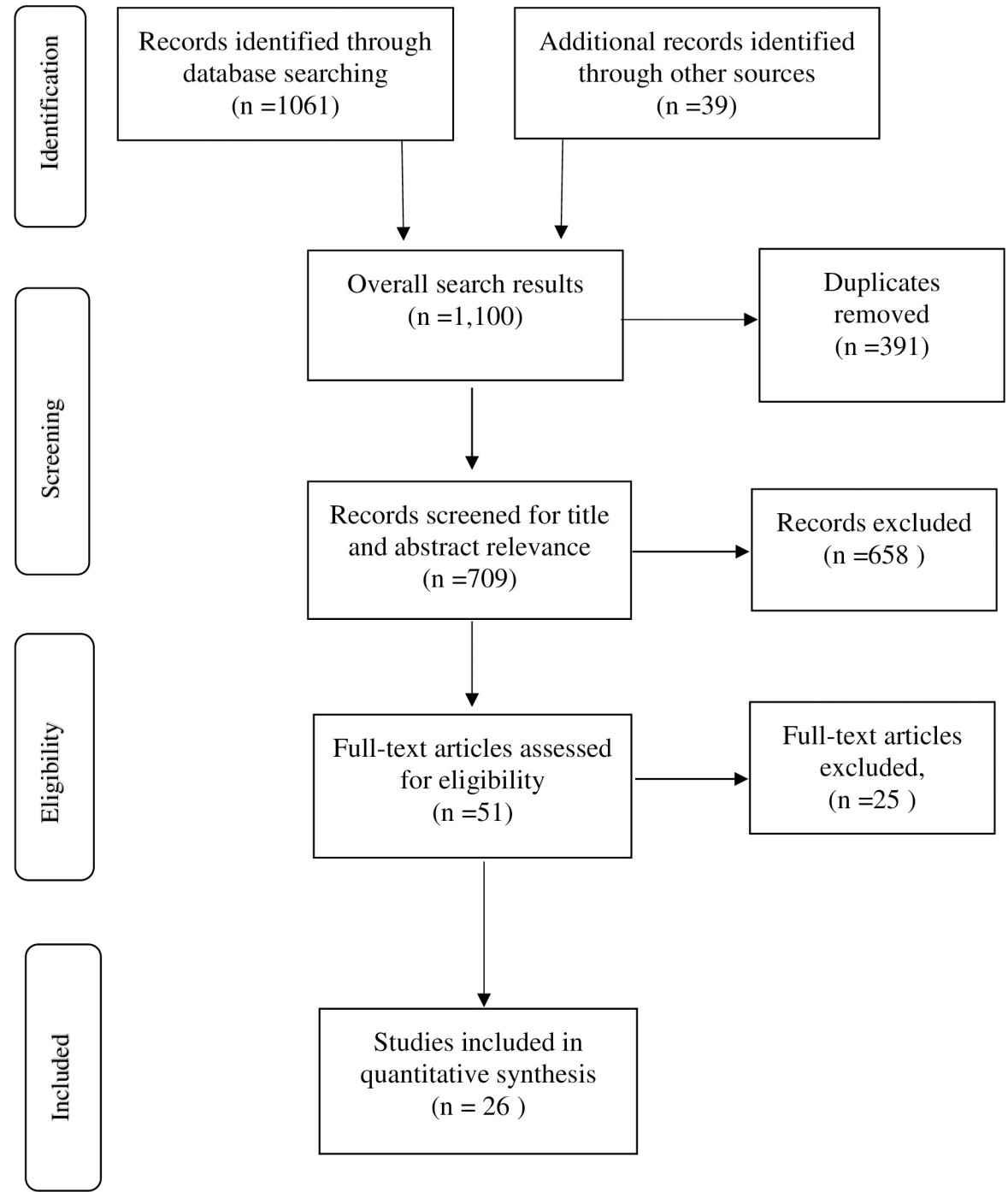

**Fig 1. PRISMA 2009 flow diagram.**

sexual and reproductive health services among PWDs. Using the constant comparison approach, points of variation or convergence in the eligible studies were highlighted to derive key thematic and sub-thematic barriers. Two authors were involved in this process and where there was a disagreement, a third review author was consulted.

**Table 1. Characteristics of included studies and key barriers facing PWDs in accessing SRH in sub-Sharan Africa.**

| Lead Author, Year | Country | Study Aim/Objective | Study Design | Participants/ Sample | Disability | Data collection methods | Barriers |
|---|---|---|---|---|---|---|---|
| Ganle et al, 2016 | Ghana | To explore the challenges women with disabilities encounter in accessing and using institutional maternal healthcare services in Ghana | qualitative study | 72 purposively sampled women | physical, visual, and hearing impairments | Semi-structured in-depth interviews | 1. Transportation difficulties to access skilled care 2. Lack of access to unfriendly physical health infrastructure 3. Healthcare providers' insensitivity and lack of knowledge about the maternity care needs of women with disability, 4. Negative attitudes of service providers, 5. Wrong perception that women with disability should be asexual 6. Health information that lacks specificity in terms of addressing the special maternity care needs of women with disability |
| Mprah et al, 2017 | Ghana | To provide more insights into the overall SRH needs of deaf people in Ghana | mixed methods | • 179 participants • 26 focus group participants • 152survey respondents • 1male key informant | Deaf (hearing impairment) | • questionnaire • focus group discussions • interviews | 1. Lack of familiarity with pregnancy prevention methods. 2. Staff at SRH centres not sensitive to the needs of deaf people 3. Limited knowledge on practices that prevent STIs/STDs |
| Mprah, 2013 | Ghana | To provide insights into factors that influence the acquisition, accessibility, and utilisation of Sexual and Reproductive Health (SRH) information and services by deaf people who communicate using Ghanaian Sign Language (GSL) | Qualitative study | 26 participants in 3 focus groups, a key informant | Deaf (hearing impairment) | focus groups discussions, a key informant, Review of documents and observations | 1. Poor quality of interpretation services 2. limited time available for consultations with health workers 3. Lack of privacy at health centres compel some deaf people to withhold information about their health 4. low literacy rate among deaf people affects access to information on SRH issues 5. Inadequate knowledge about deaf people by health affect effective interaction and communication 6. negative attitudes of health staff towards deaf people 7. non-use of deaf people's preferred means of communication (absence of sign language interpreters at SRH centres) 8. services that are not customised to their needs |
| Tun et al, 2016 | Ghana, Uganda and Zambia | To explore how the barriers faced by persons with disabilities living with HIV impede their ability to access HIV-related services and manage their disease. | qualitative study | 76 participants (41 females; 35 males) | Hearing Visual Physical | FGD | 1. Challenges in getting to the health facilities, 2. Lack of information about HIV and testing 3. HIV- and disability-related stigma 4. Delays in testing for HIV 5. Lack of disability-friendly educational materials and Lack of sign interpreters 6. Stigmatizing treatment by providers and other patients 7. Lack of skills to provide tailored services to persons with disabilities living with HIV 8. Physically inaccessible infrastructure |

*(Continued)*

**Table 1.** (Continued)

| Lead Author, Year | Country | Study Aim/Objective | Study Design | Participants/ Sample | Disability | Data collection methods | Barriers |
|---|---|---|---|---|---|---|---|
| Schenk et al 2020 | Uganda, Zambia, and Ghana | To explores access to and use of HIV information and services among persons with disabilities. | Qualitative study | 21 KIIs with government officials 44 FGDs among persons with disabilities | Sensory, physical) and caregivers for persons with intellectual and developmental disabilities | key informant interviews and focus group discussions | 1. Lack of information, misinformation and community beliefs about sexual activity among persons with disabilities 2. Literacy challenges knowledge about HIV was limited and often associated with illiteracy 3. Vulnerability to abuse 4. Poor attitude of healthcare providers to persons with disability 5. complexity of stigma across multiple layers stigmatising cultural beliefs; |
| Tanabe et al, 2015 | Kenya, Nepal, and Uganda | To explore the risks, needs, and barriers for refugees with disabilities to access SRH services, and the practical ways in which these challenges could be addressed | Qualitative study | • Women • Men • Adolescent girls and boys (15–19 yrs) • Caregivers and family members | Physical Intellectual Sensory Mental impairments | FGD interviews | 1. Lack of respect by providers 2. Pregnant women with disabilities were often discriminated against by providers 3. Marital status was a large factor 4. Risks of sexual violence among persons with intellectual impairments 5. Limited awareness around SRH 6. Negative and disrespectful provider attitudes (Kenya and Uganda) 7. Long wait times (Kenya and Uganda) 8. Costs to seeking care (Uganda), 9. Refugee status (Uganda), 10. Communication with providers (all three sites) Caregiver and community attitudes (Uganda) 11. Lack of transportation (Kenya and Uganda) 12. Limited accessibility (all three sites). 13. lack of translation for both spoken and sign language 14. lack of transportation to health facilities; 15. limited wheelchair availability at the referral hospital; stock-outs of medicines; 16. lack of money to pay health providers 17. discrimination of unmarried woman with disabilities 18. women with disabilities were observed to have less stable relationships and were subsequently caring for children without a partner |
| Van Rooy & Mufune, 2014 | Namibia | to investigate the experiences of people with disabilities (PWD) regarding issues of sexuality and HIV/AIDS | qualitative study | Senior government officials (5) Females with disabilities (5) and a group of males and females with disabilities (12). | Visual disabilities 5 (18.5%) Physical disabilities 11 (40.7%) Hearing disability 3 (11.1%) | Key informant interviews and focused group interviews (FGDs). | 1. Negative public attitudes towards PWD who engage in sex and pregnant PWD 2. Materials on HIV/AIDS education is not written in Braille or otherwise fail to consider the different disabilities 3. Problems accessing reproductive health services because of the negative attitudes of healthcare providers |
| Rugoho & Maphosa, 2017 | Zimbabwe | To explore the challenges faced by women with disabilities in accessing sexual and reproductive health in Zimbabwe | qualitative study | 23 participants 18 to 45 years. | physical disabilities, visually impaired, deaf and stammering | In-depth interviews | 1. Negative perceptions of health personnel towards people with disabilities, 2. Disability unfriendly infrastructure at health facilities 3. Absence of trained personnel for people with disabilities (sign language) |

(*Continued*)

**Table 1.** (Continued)

| Lead Author, Year | Country | Study Aim/Objective | Study Design | Participants/ Sample | Disability | Data collection methods | Barriers |
|---|---|---|---|---|---|---|---|
| Peta, 2017 | Zimbabwe | To elucidate the childbearing experiences and aspirations of women with disability in Zimbabwe | qualitative study | | mental, physical and sensory disabilities, | In-depth interviews | 1. Inability of health-care staff to use sign language to communicate with her in the appropriate language<br>2. Discrimination against women with disability due to cultural understanding of disability, which associates disability with evil spirits, taboos and witch-craft<br>3. Self-stigmatisation on the grounds of disability<br>4. Lack of support of women with disability in reproductive health clinics, in relation to issues of contraceptives<br>5. Women with disability may be denied access to sexual and reproductive health information |
| Burke et al, 2017 | Senegal | To understand what barriers and enablers young people with disabilities experience when accessing SRH services | qualitative study | | physical or sensory (visual or hearing) impairment | FGDs and in-depth interviews | 1. Low knowledge about, and use of, SRH services<br>2. Multiple cases of rape amongst women with hearing impairments.<br>3. Key barriers to SRH services were financial barriers<br>4. Provider attitudes and accessibility (related to their disability) |
| Ahumuza et al. 2014 | Uganda | To explore the challenges faced by male and female persons with physical disabilities in accessing SRH services in Kampala, Uganda | qualitative study | 40 PWPDs 10 PWPDs' representatives, staff of agencies supporting PWPDs health workers | | in-depth interviews | 1. Negative attitudes of service providers<br>2. Long queues at health facilities,<br>3. Distant health facilities,<br>4. High costs of services involved,<br>5. Unfriendly physical structures<br>6. the Negative public perception that PWPDs should be asexual |
| Mulumba et al. 2014 | Uganda | To gain a deeper understanding of the perceptions and experiences of older people and persons with disability on accessing public healthcare and inter-related social services | qualitative study | | Hearing impairment Physical impairment | focus group discussions and key informant interviews | 1. Lack of adjustable hospital beds in delivery wards for women giving birth,<br>2. Communication barriers between physicians and their patients with disabilities |
| Apolot et al, 2019 | Uganda | The study explores the maternal and newborn health related needs of women with walking disabilities in Kibuku District Uganda. | qualitative study | 4 | walking disabilities | In-depth Interviews | 1. Psychosocial needs during pregnancy, delivery and the postnatal period. These included acceptance by: partners, families, communities and health workers.<br>2. Transport-related needs. The suitability of transport, the difficulty in finding transport and the high costs involved.<br>3. infrastructural and special service needs at the health faclities: lower examination and delivery beds, seats, ramps, and sanitary facilities,<br>4. Respondents also expressed a need for special outreach services for antenatal and postnatal care.<br>5. Long waiting time for healthcare delivery |

(*Continued*)

**Table 1.** (Continued)

| Lead Author, Year | Country | Study Aim/Objective | Study Design | Participants/ Sample | Disability | Data collection methods | Barriers |
|---|---|---|---|---|---|---|---|
| Yousafzai et al, 2005 | Uganda & Rwanda | To determine factors which may increase vulnerability of disabled adolescents to HIV infection and/or inappropriate access to HIV related services | qualitative study | disabled adolescents, non-disabled adolescents, parents, teachers, members of disabled people's organisations and representatives of HIV/AIDS organisations | | focus group discussions and semi-structured interviews | 1. Inaccessible information<br>2. Inappropriate teaching techniques<br>3. Poverty<br>4. Stigma<br>5. Difficulties in accessing health facilities<br>6. Negative attitudes towards people with disability in relation to HIV testing<br>7. Ignorance about disability and sexuality by both disabled adolescents and non-disabled people<br>8. Low self-esteem and issues of self-efficacy affect their control of safer sexual relationships<br>9. Rape of people with disabilities<br>10. Physical inaccessibility<br>11. Lack of privacy<br>12. Negative attitudes<br>13. Perceptions of low risk for HIV infection |
| Gichane et al, 2017 | South Africa | To describe and compare the pregnancy outcomes and maternity service use of a sample of signing deaf women of child-bearing age in Cape Town to the population of the Western Cape of South Africa. | Survey | 42 women | Deaf | Questionnaire | 1. Inadequate interpretation of maternal health services<br>2. Reported experience of mistreatment from hospital staff |
| Mavuso and Maharaj, 2015 | South Africa | To gain insight into the experiences and perspectives of PWD regarding their access to sexual and reproductive health services | qualitative study | 16 participants 10 women and 6 men of reproductive age. | physical, visual and hearing disabilities | in-depth interviews | 1. Societal discriminatory attitudes towards PWD.<br>2. Sexual exploitation, thereby increasing their vulnerability to STIs including HIV and AIDS.<br>3. No information on sexual and reproductive health available in alternative formats such as braille, enlarged print or audio compact disks<br>4. Experiences poor treatment at health facilities<br>5. Health facilities that offer sexual and reproductive services are in difficult to access areas |
| Oladunni, 2012 | Nigeria | To determine access of adolescents with disabilities to Sexuality Information in Osun state | Mixed methods | 140 79 males 61 females | physical disabilities | questionnaires and interview | 1. Poor access to sexuality information<br>2. Low capacity to manage sexual difficulties and engagement in unsafe sex<br>3. Non-recognition of sexual and reproductive needs and rights of adolescents with disabilities on existing national curriculum<br>4. Inadequate capacity of educators on the topic of sexuality and disability.<br>5. Absence of relevant curriculum, teaching materials and other resources that can enhance effective teaching and learning of sexuality education among PWDs |
| Oladunni, 2012 | Nigeria | To investigate the sexual behavior and practices of adolescents with disabilities in Osun State. | Cross sectional study | 140 (79 males & 61 females) | physical disabilities | questionnaires | 1. Poor knowledge of sexuality issues<br>2. Lack of access to HIV counseling and testing<br>3. sexual assault; rape and molestation |

(*Continued*)

**Table 1.** (Continued)

| Lead Author, Year | Country | Study Aim/Objective | Study Design | Participants/Sample | Disability | Data collection methods | Barriers |
|---|---|---|---|---|---|---|---|
| Smith et al, 2004 | Zambia | To identify whether there are currently any physical, social and/or attitudinal barriers to women's with disability use of reproductive health services | Qualitative study | 24 purposely selected women with disabilities and with 25 safe motherhood/reproductive public-sector health service providers. | Physical disability | In-depth interviews | 1. Negative traditional beliefs towards PWDs 2. Expectations of poor care & bad attitudes. 3. Attitudes of others. 4. Fear of bad reception, complications, & caesarean section 5. Unnecessary referral. 6. Ignorance / lack of knowledge about disability 7. Poverty 8. Transport system. 9. Exclusion from health education & community activity Traditional beliefs. 10. Lack of information about disability 11. Distance to health facilities. 12. Lack of mobility assistive devices 13. Inaccessible minibuses |
| Parsons et al, 2015 | Zambia | To explore the experiences of persons living with disabilities in Lusaka, Zambia who became HIV-positive | Qualitative study | 32 participants (21 PWD/HIV+ and 11 key informants | physical, visual hearing intellectual | Inductive thematic analysis | 1. Stories of stigma in the clinical encounter 2. Stories of stigma within the community 3. Accounts of internalized stigma |
| Nixon et al, 2014 | Zambia | To explore perceptions and experiences of HIV-related health services for PWDs who are also living with HIV in Lusaka, Zambia. | Qualitative study | 21 PWDs who had become HIV-positive, and 11 people working in HIV and/or disability | physical, hearing, visual intellectual impairments | in-depth, semi-structured, one-on-one interviews | 1. Disability-related discrimination during access to HIV services, 2. Ccommunication barriers 3. Concerns with confidentiality 4. Movement and mobility challenges related to seeking care and collecting antiretroviral therapy |
| Bremer et al, 2009 | Cameroon | To investigate the reproductive health experiences among women with physical disabilities in the Northwest Region of Cameroon | Qualitative study | 8 participants | mobility (physical) impairments | semi-structured key informant interviews | 1. Healthcare workers were not knowledgeable in disability issues, nor sensitive to their needs 2. Most healthcare centers were inaccessible, 3. Taxis unwilling to carry them to access services 4. physical inaccessibility 5. financial barriers to reproductive health services 6. Inability to afford cost of transportation to health centers 7. Feeling of isolation due to mobility impairment or stigma |
| DeBeaudrap et al, 2019 | Cameroon | To examine to what extent socioeconomic consequences of disability contribute to poorer access to sexual and reproductive health (SRH) services for Cameroonian with disabilities and how these outcomes vary with disabilities characteristics and gender | Cross-sectional study | 310 persons with disability and another 310 without disability were included in the analysis. | Physical Visual hearing | Face-to-face structured interviews | 1. Limited access to SRH by women with disability 2. Restricted access to education affects their low use of family planning and HIV testing services |

(*Continued*)

**Table 1.** (Continued)

| Lead Author, Year | Country | Study Aim/Objective | Study Design | Participants/ Sample | Disability | Data collection methods | Barriers |
|---|---|---|---|---|---|---|---|
| Beyene et al, 2019 | Ethiopia | to assess modern contraceptive use and associated factors among women with disabilities in Gondar city, Ethiopia | cross-sectional study | 267 reproductive age women with disabilities | hearing, visual and limb defects (physical impairments) | house-to- house interview. | 1. Existing family planning service delivery points were not accessible, difficult to access 2. health professionals'attitude was not good 3. Educational status of respondents was found to be a significant predictor for modern contraceptive use. |
| Tefera et al, 2017 | Ethiopia | The grace of motherhood: disabled women contending with societal denial of intimacy, pregnancy, and motherhood in Ethiopia. | Qualitative study | 13 participants | physical or visual disabilities | In-depth semi-structured interviews personal observations | 1. Negative societal attitudes toward women with disabilities regarding relationship, pregnancy, and child-rearing. 2. Accessibility of health centers 3. ignorance and negative attitudes of the physicians |
| Yimer & Modiba, 2019 | Ethiopia | To determine the knowledge and practice level on modern contraceptive methods among blind and deaf women about in Addis Ababa City, Ethiopia. | Mixed methods design | 326 cases (164 deaf and 162 blind women). | blind and deaf women | Questionnaire Key informant interviews and personal observation | 1. level of comprehensive knowledge on modern contraceptive methods was lacking 2. numerous misunderstandings and myths about disability and SRH. 3. lack of appropriate information communication means and modes which target persons with sensory impairments. there were no any written, visual or audio materials at assessed health facilities for persons with sensory impairments 4. Lack of capacity of service providers as they have very little training in relation to disability and limited access to the resources that would enable them to provide a disability inclusive SRH services. |

### Role of the funding source

The funder of the study had no role in study design, data collection, data analysis, data interpretation, or writing of the report. The corresponding author had full access to all the data in the study and had final responsibility for the decision to submit for publication.

## Results

### Study characteristics

In all, 1061 articles were retrieved from five electronic databases comprising Medline Complete = 472, CINAHL Complete = 372, PsycINFO = 220, Embase = 81, and Web of Science Core Collection = 23 (**see Fig 1**). An additional 39 articles were retrieved from other sources including reference list of eligible studies. Some 391duplicates were removed and the remaining screened from title and abstracts for relevance. Further, 658 studies were excluded after title and abstract screening leaving 51 articles for which full text were obtained. At the end, 26 articles met the inclusion criteria and were retained for analysis. **Table 1** presents the characteristics of the eligible studies.

Of the 26 eligible studies, 19 were qualitative in design [7, 10, 11, 18–20, 31–43]; four used a survey design [44–47]; and three used mixed methods design [48–50]. The 26 studies were reported from twelve African countries: Uganda (seven studies) [7, 10, 31, 32, 34, 37, 39];

**Table 2. Summary of key barriers PWDs face in accessing SRH services and information in sub-Saharan Africa.**

| Level at which barriers are encountered | Specific barrier type | Number of studies reporting specific barrier |
|---|---|---|
| National level | Unfriendly legal environment and policies towards SRH issues for PWDs | 1 |
| | Unfriendly/lack of appropriate public transportation services | 8 |
| | Limited education opportunities for PWD on SRH issues | 2 |
| Individual level | Sex (gender) | 5 |
| | Socio-cultural/religious beliefs and practices | 9 |
| | Refugee status | 1 |
| | Low literacy rates among disabled people | 2 |
| | Lack of knowledge/ignorance (awareness) on SRH issues | 7 |
| | Communication barrier | 9 |
| | Lack of self-efficacy | 3 |
| Community level | Negative public attitudes towards PWDs' sexuality issues | 8 |
| | Stigma and discrimination against disabled patients | 5 |
| | Sexual violence and abuse at the community level | 6 |
| | Lack of community or family support network | 11 |
| Health system/ institutional level | Poor interpersonal relationships | 1 |
| | Limited/lack of knowledge/capacity on PWD SRH issues | 10 |
| | Insensitivity/negative attitudes | 16 |
| | Discrimination | 7 |
| | Limited consultation time | 1 |
| | Inaccessibility or lack of SRH information/resources | 6 |
| | Low staff capacity/numbers | 1 |
| | Lack of access to HIV counseling and testing | 1 |
| | Lack of adaptation of health information to suit PWDs | 5 |
| | Lack of privacy and confidential services | 8 |
| | Lack of translators/sign language specialists | 6 |
| | Limited availability of special outreach services for antenatal and postnatal care targeting persons with disabilities. | 1 |
| | Stock outs of medicine/medical services | 2 |
| | Lack of wheelchairs/mobility aids | 2 |
| | Unfriendly HIV/aids education materials | 1 |
| | Long waiting times | 6 |
| | Disability unfriendly physical infrastructure | 8 |
| | Lack of adjustable beds for delivery | 2 |
| Economic level | Cost of service | 4 |
| | Financial constraints | 7 |
| **Number of specific barriers identified** | | 33 |

Ghana (five studies) [11, 32, 39, 49, 51], Zambia (five studies) [18, 32, 36, 38, 39]; Zimbabwe (two studies) [19, 43]; South Africa (two studies) [42, 52]; Kenya (one study) [31]; Namibia (one study) [53]; Senegal (one study) [20]; Nigeria (two studies) [45, 48]; Rwanda (one study) [54]; Cameroon (two studies) [46, 55]; and Ethiopia (three studies) [35, 47, 50].

Fourteen out of the 19 eligible qualitative studies were rated as high quality [7, 10, 11, 19, 20, 32, 35–39, 42, 51, 55]; five studies were medium to high quality [18, 31, 43, 53, 54] (**see S2 File**). Most of the studies did not provide adequate information on how participants were recruited. Except one study [55], none of the studies provided information on the relationship

between researcher and participants (reflexivity). Only one quantitative study was rated high [46] and the rest rated medium to low [45, 47, 52] **(see S3 File)**. Lastly, the mixed methods studies [45, 49, 50] were assessed as average or medium quality **(see S4 File)**.

## Synthesis of barriers to SRH services among PWDs

Overall, 33 specific barriers hindering access to SRH information and services among PWDs were identified. These barriers were further categorised into five levels: national, health system/institutional, individual, community and economic (**see Table 2**). Nine out of the 26 eligible studies identified three different set of barriers that PWDs face accessing SRH services at the national level [11, 18, 31, 32, 42, 43, 53]. These were unfriendly/lack of appropriate public transportation services [11, 18, 31, 37, 42, 53, 55], and limited education opportunities for PWD on SRH issues [35, 37].

Eighteen different barriers were identified under the healthcare system/institutional level. These included lack of SRH information/resources at healthcare settings [19, 32, 42, 43, 48, 50]; low staff capacity/numbers [38]; lack of adaptation of health information to suit PWDs [11, 32, 47, 54, 55]; lack of privacy and confidential services at the point of access [20, 32, 36, 42, 43, 51, 53, 54]; lack of translators/sign language specialists [19, 31, 32, 38, 47, 49]; frequent stock outs of essential SRH commodities [31, 53]; lack of wheel chairs and mobility aids at the facility level [31, 37]; unfriendly HIV/AIDS education materials [53]; longer waiting times [7, 31, 32, 37, 53]; and disability unfriendly physical healthcare infrastructure [7, 31, 32, 35, 37, 42, 43, 55]. Other barriers included poor interpersonal relationships between PWDs and healthcare providers [37]; lack of knowledge or limited capacity of staff on PWDs SRH issues [7, 11, 18, 32, 42, 48, 50, 51, 53, 55]; and insensitivity/negative attitudes of healthcare staff towards PWDs [7, 11, 19, 20, 31, 32, 35, 39, 42, 43, 47, 49, 51–53, 55]. The rest comprised discrimination against PWDs by healthcare providers [19, 20, 31, 32, 36, 42, 54]; limited consultation time [48]; lack of access to HIV counseling and testing [45]; and limited availability of special outreach services for antenatal and postnatal care targeting persons with disabilities [37].

Individual level barriers were the next most predominant. Sixteen studies reported seven different individual level barriers. These included gender inequalities [31, 32, 35, 46, 53]; negative socio-cultural/religious beliefs and practices [35, 38, 39]; refugee status [31]; low literacy rates among PWDs [48]; lack of knowledge/awareness on SRH issues [20, 31, 32, 39, 45, 49, 55]; communication barriers [11, 20, 36, 39, 49, 50, 52–54]; and lack of self-efficacy [18, 38, 48].

Four different sets of community level barriers were identified in fourteen studies. These included negative public attitudes towards PWDs and their sexuality issues [7, 11, 18–20, 31, 37, 53]; stigma and discrimination against disabled patients/clients [32, 35, 37, 38, 54]; sexual violence and abuse at the community level [20, 31, 39, 45, 53, 54]; and lack of community or family support networks to enable PWDs access SRH services and information [11, 19, 20, 32, 35, 37, 39, 49, 53–55].

Lastly, two main economic barriers emerged from review of eligible studies. These included unaffordability of SRH services and information [18, 37, 49, 54]; and general financial and resource poverty which hinder PWDs' access to SRH services and information [7, 20, 31, 32, 35, 38, 51].

Overall, these barriers can be categorised into demand-side and supply-side barriers. For example, demand-side barriers relate PWDs' lack of self-esteem, high level of illiteracy rates or lack of education, community and family level stigmatizatio which undermine access to SRH services, and lack of access to financial resources to access SRH services [11, 19, 36, 53]. Supply-side barriers could include discrimination against PWDs. at health facilities by healthcare

workers, disability unfriendly healthcare facilities, lack of disability-friendly wash rooms delivery/labour wards in healthcare facilities, and communication barriers between healthcare providers and PWDs [7, 11, 20, 32, 42]. It is also important to note that whilst some of the barriers are peculiar to persons with specific types of disability, many other barriers are faced by the general population. For example, studies by Mphrah [49] and Gichane [52] show that persons with hearing difficulties (deaf) faced particular types of barriers including poor quality sign language interpretation services and inadequate knowledge about deaf people. These barriers specifically prevent them from accessing SRH services in healthcare facilities due to lack of effective interaction and communication systems. However, barriers such as lack of SRH information/resources at healthcare settings, low staff capacity/numbers, lack of privacy and confidential services at the point of access apply to the general public.

## Discussion

This paper appraised evidence on the barriers persons with disabilities face in accessing sexual and reproductive health information and services in sub-Saharan Africa. Five levels of barriers covering a total of 33 specific barriers were identified after pooling studies. The barriers identified comprised broader national level, healthcare system level, individual level, community level, and economic barriers. Many of the specific barriers identified however overlapped across studies, clinical settings and geographical contexts. Overall, these findings are largely consistent with previous related research on the barriers to sexual and reproductive health services in other contexts outside Africa [56–58], and on access to general healthcare services [59, 60]. For example, the present review identified lack of education and knowledge on sexual and reproductive health services and information, poor treatment of PWDs by healthcare workers, and disability unfriendly healthcare facilities and services. Studies in Nepal [61], UK [56] and India [58] have reported similar findings. In a previous review covering low and middle-income countries, barriers to general healthcare services that persons with disability faced included lack of information, limited mobility, stigmatization, and negative and poor staff attitude [62]. This congruency underscores the fact that some of the barriers PWDs face in accessing sexual and reproductive health information and services may be global in nature and thus a well-concerted global response is needed.

The findings of this review have implications for policy, practice and future research. From this review, it is clear that PWDs face myriad of both demand and supply side barriers to accessing sexual and reproductive health services and rights in sub-Saharan Africa. If the Sustainable Development Goals' 3 objective of universal and/ or equitable access to skilled and comprehensive sexual, reproductive and maternal health services is to be attained in sub-Saharan Africa, urgent context-specific policy actions and disability-appropriate interventions are needed to address the barriers identified in this review. Barriers such as maltreatment of PWDs by healthcare professionals undermine the rights of PWDs to access sexual and reproductive health and rights. This requires policy and management attention to train healthcare providers on interpersonal communication skills and relationships. This could enable healthcare providers deliver healthcare services with high level of sensitivity and fairness. Similarly, limited availability of access ramps posed a great deterrent to access to services among PWDs [11, 32, 38]. This requires management of health facilities to ensure adequate provision of access ramps to facilitate better access for persons with physical disabilities. Also, some healthcare workers lacked the requisite professional skills to deal with PWDs. Therefore, we advocate further and regular training of healthcare workers on how to provide sexual and reproductive healthcare information and services to PWDs in a respectful and non-judgmental manner. Specific courses on providing care to disabled persons should be incorporated into the

curricular of health training institutions. Training manuals on this topic should also be made available, with such trainings segregated according to the types of disability, their culture and unique healthcare needs. This will ensure such trainings are context-specific and seek to identify and address the needs of specific disability groups.

Another issue relates to the lack of support from communities and families, which is fueled by misperceptions about PWDs and their sexuality. These misperceptions and beliefs are borne out of poor understanding of disability as well as lack of awareness about the sexuality and sex life of PWDs. It is therefore important that public educational interventions are designed and implemented to demystify such prevailing beliefs and practices, improve public understanding of the sexual and reproductive health needs of PWDs and ways the public could support PWDs to fully enjoy their fundamental human rights in relation to safe and satisfying sexual life. In a similar vein, PWDs should be educated more on issues relating to their sexual rights and access to sexual and reproductive health services. This is important to overcome lack of awareness on sexual and reproductive health issues and lack of self-efficacy among PWDs. Indeed, insights from this review could be used to develop an evidence-based implementation strategy on how to address access barriers at the various levels: national level, institutional or health system context, economic context, individual PWDs, community and family level contexts. This could, for example, include developing training guidelines and tools for instruction at the various health training institutions for healthcare providers.

Finally, although different types of disabilities exist, the review showed that there is relatively more scholarly attention on particular types of disabilities compared to others. The studies reviewed reported more on hearing/speech impairments, visual impairment and physical disability and less on other types of disability such as mental or intellectual disability. Further studies are thus required to bring to light the barriers faced by people with these types of disabilities. The international literature on PWDs suggests that females face more barriers in general compared to their male counterparts. This was however not clearly articulated in this review. We suggest that future research delves further into the gender-based barriers PWDs face in accessing sexual and reproductive health services. This will ensure that existing health programmes and interventions are sensitive to, and addresses, the unique needs of both females and males.

## Study limitations

A number of limitations should be noted in this review. First, a number of barriers were identified as hindering access to sexual and reproductive health services among PWDs. However, this review did not indicate the extent to which such barriers interact or influence one another, and the ways in which such interactions determine access. Second, although a comprehensive search strategy was designed and conducted in five key biomedical and health sciences databases using broadly defined search terms, keywords and queries to identify and synthesize findings relevant to PWDs' access to sexual and reproductive health services, only 26 articles were considered eligible for review. It is likely that some relevant articles were still missed due to language restriction. The search approach was restricted to only English language publications and it is plausible this resulted in the exclusion of eligible studies published in languages other than English. The search was also confined to only peer-reviewed journal articles, thus relevant editorials, theses, conference presentations which may have extended the depth of evidence on the topic were excluded. Third, this review adopted a multi-study design approach in enlisting eligible articles. Consequently, a meta-analysis was not permissible to assess the pooled effects of the barriers to accessing sexual reproductive health services. Another limitation of the study was the inability to establish confidence on the weight of the barriers to

accessing SRH services by PWDs based on the overall frequencies. The GRADE tool may be considered in future reviews to establish the strength of the evidence presented in this review. Nevertheless, some strengths of the present review are notable. To the best of our knowledge, the present review presents the first attempt to comprehensively and systematically identify and synthesize both qualitative and quantitative studies on the barriers PWDs face in accessing sexual and reproductive health information and services in sub-Saharan Africa. A further strength of this review is that the search followed the PRISMA protocol, an internationally recognized best practice methodology in undertaking systematic reviews.

## Conclusion

The present study was conducted to document and appraise evidence on the barriers persons with disabilities face in accessing sexual and reproductive health services in sub-Saharan Africa. The review found a myriad of barriers faced by PWD in their attempt to access SRH services, which have been categorized into five levels: broader national level barriers; health-care system/institutional barriers; individual level barriers; community level barriers; and economic barriers. The barriers were also specific to particular forms of disabilities and varied across different SRH services. Efforts by policy makers to improve access to SRH services by PWD need to pay attention to these contextualized barriers.

## Supporting information

**S1 Checklist. PRISMA 2009 checklist.**
(DOC)

**S1 File. Medline search strategy.**
(DOCX)

**S2 File. Quality assessment results of qualitative included studies (CASP).**
(DOCX)

**S3 File. Quality assessment results of quantitative included studies (CEBM).**
(DOCX)

**S4 File. Quality assessment results of mixed methods design using both CASP & CEBM tools.**
(DOCX)

## Author Contributions

**Conceptualization:** John Kuumuori Ganle.

**Data curation:** John Kuumuori Ganle, Leonard Baatiema.

**Formal analysis:** John Kuumuori Ganle, Leonard Baatiema.

**Investigation:** John Kuumuori Ganle, Leonard Baatiema.

**Methodology:** John Kuumuori Ganle, Leonard Baatiema, Reginald Quansah, Anthony Danso-Appiah.

**Supervision:** Anthony Danso-Appiah.

**Validation:** Reginald Quansah, Anthony Danso-Appiah.

**Writing – original draft:** John Kuumuori Ganle, Leonard Baatiema, Reginald Quansah.

**Writing – review & editing:** John Kuumuori Ganle, Reginald Quansah, Anthony Danso-Appiah.

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
