## [Decision Letter · Decision Letter 0]

15 Jun 2020

PONE-D-20-07319

Barriers facing persons with disability in accessing sexual and reproductive health services in Sub-Saharan Africa: a systematic review

PLOS ONE

Dear Dr. Ganle,

Thank you for submitting your manuscript to PLOS ONE. After careful consideration, we feel that it has merit but does not fully meet PLOS ONE’s publication criteria as it currently stands. Therefore, we invite you to submit a revised version of the manuscript that addresses the points raised during the review process.

We look forward to receiving your revised manuscript.

Kind regards,

Joshua Amo-Adjei, Ph.D

Academic Editor

PLOS ONE

Journal Requirements:

2. Please include captions for your Supporting Information files at the end of your manuscript, and update any in-text citations to match accordingly. Please see our Supporting Information guidelines for more information: http://journals.plos.org/plosone/s/supporting-information

Reviewers' comments:

Reviewer's Responses to Questions

**Comments to the Author**

1. Is the manuscript technically sound, and do the data support the conclusions?

Reviewer #1: Yes

Reviewer #2: Yes

Reviewer #3: Yes

2. Has the statistical analysis been performed appropriately and rigorously? 

Reviewer #1: No

Reviewer #2: N/A

Reviewer #3: Yes

3. Have the authors made all data underlying the findings in their manuscript fully available?

Reviewer #1: Yes

Reviewer #2: No

Reviewer #3: Yes

4. Is the manuscript presented in an intelligible fashion and written in standard English?

Reviewer #1: Yes

Reviewer #2: Yes

Reviewer #3: Yes

5. Review Comments to the Author

Reviewer #1: I would like to commend the authors for writing an interesting piece of review. I believe this is a topic of interest to reproductive health researchers, policy-makers and practitioners, and I would recommend that it be published. However, some changes need to be effected; or rather, considered to strengthen the review before it is published.

1. Authors have stated the rationale for this review vividly and it is within the context of what is already known. However, the statement that the evidence has not been systematically documented in Sub-Saharan Africa, to me appears weak and the statement has been emphasized every section of the text. it should be reworked on based on factors to be studied and should appear only in the introduction and abstract section. What is reason for consolidating the papers? The fact that the papers have not been systematically documented does not translate to a knowledge gap. Can you come up with an appropriate reason for your review?

2. It would help readers if you provided a nice and clear definition of 'disabilities' the Introduction and also, a sentence or two outlining its impact globally, i.e. why/how access to sexual and reproductive health services is a problem/important for this particular group?

3. Methods: It is no longer recommended that searching and screening be restricted to English-only as this can introduce language bias. Your statement of eligibility “with only English Language restriction” is already present bias. Majority of abstracts will provide an English translation for title and abstract screening; and there are online translation programs (e.g. Google Translate) that will normally provide an understandable English translation of full text reports. So your criteria for language should read 'must be in English or able to translate into understandable English'

4. Search strategy, should be detailed, explicable and with relevant key words/search terms. I tried to google and the feedback with information retrieved were overwhelming. Come up with a sub-heading detailing the inclusion and exclusion criteria used

5. It is nice to see that you tried to frame your research question in PICO format however, without the specific question to help you address those concern, it would be like speculation. Please sentence of a sentence of a research question that begin…”what is the evidence that…..

6. Study selection: You do not need to mention the initial of the authors in the text (same with data extraction). There is a section of author role dedicated by the journal for this. However, you need to explain chronological event involved in the study selection and that disagreement between two reviewers to be resolved by a third reviewer, if you invite two or more reviewers as in between a further disagreement may ensue and the circus continues….

7. How would you manage issues of missing data? Please explain

8. Now that improvement in the assessment of evidence is currently in the works. It would be important to consider use of GRADE tool for quality ratings. I observed that you focused on both qualitative and quantitative data, and stated that Most of the studies did not provide adequate information on how participants were recruited. In addition, that the quantitative studies had a medium to low quality rating. As such methods may differs in terms of quality of evidence, thus GRADE should take care of risk of bias across studies, publication bias among others. Note: I was not able to access supplementary file 3 and 4 for verification.

Reviewer #2: Thanks to the authors for this interesting paper entitled “Barriers facing persons with disability in accessing sexual and reproductive health services in Sub-Saharan Africa: a systematic review”. Overall, I think this is an excellent piece of work and only have minor comments.

1. The authors should be consistent with the use of sub-Saharan Africa

2. Page 3 Line 71 and the entire manuscript, Please check the referencing style of PLOS ONE and reformat them accordingly

3. Page 3 Line 75 and line 124 authors have defined their abbreviation, PWDs (Persons with Disability). Please start the sentences with PWDs instead of Persons with Disability

4. The authors might wish to consider a recent study “Kumi-Kyereme, A., Seidu, A. A., & Darteh, E. K. M (2020). Factors Contributing to Challenges in Accessing Sexual and Reproductive Health Services Among Young People with Disabilities in Ghana. Global Social Welfare, 1-10” at their literature review section or background. https://www.researchgate.net/publication/341578212_Factors_Contributing_to_Challenges_in_Accessing_Sexual_and_Reproductive_Health_Services_Among_Young_People_with_Disabilities_in_Ghana

5. Page 4 Line 127 what informed the authors to use age 15 as the starting point?

6. Page 6 Line 174, kindly replace “reviews” with “reviewers”

7. Page 6 Line 194 specify these two authors

8. Page 7 Line 210 to 216 and line 233-237 please take a critical look at the use of comma (,)

9. Page 9 Line 274-275 kindly replace developing country with low and middle-income countries

10. Page 9 between line 279 and 280 consider creating a sub-section “Strengths and Limitations”

11. Similarly, Kindly consider creating a subsection “implications for policy, practice and future research” between line 297 and 298.

12. Although the authors have presented/fused their conclusions at the “implications for policy, practice and future research” section, it will be appropriate to have a separate heading/section for the conclusion they drew from their review

Reviewer #3: This is a well written paper that addresses an important but neglected topic. The authors justified the need for the synthesis and appeared to have rigorously conducted this study. Few points of clarity that I want the authors to consider.

1) Since we just in the middle of 2020, what month was the search conducted? how many 2020 paper were included? perhaps capping the study period to 2019 will be fine, otherwise, state the month in 2020 the search ended.

2) Since this study is about SSA, I wonder why AJOL, where many African Journals are represented was not searched. If there a justification for this?

3) Authors did not discuss the limitations of this study. The study only came from 12 of over 40 SSA countries. To what extent is the study representative of the SSA?

4) Did authors reviewed international frameworks and SSA countries policies on people with disability and what was found. Policies on people with disabilities in SSA will help provide more context to the findings of the study.

6. PLOS authors have the option to publish the peer review history of their article (what does this mean?). If published, this will include your full peer review and any attached files.

Reviewer #1: Yes: Gabriel O Dida

Reviewer #2: Yes: Abdul-Aziz Seidu

Reviewer #3: Yes: Anthony Ajayi

---

## [Decision Letter · Decision Letter 1]

20 Aug 2020

Barriers facing persons with disability in accessing sexual and reproductive health services in Sub-Saharan Africa: a systematic review

PONE-D-20-07319R1

Dear Dr. Ganle,

We’re pleased to inform you that your manuscript has been judged scientifically suitable for publication and will be formally accepted for publication once it meets all outstanding technical requirements.

Kind regards,

Joshua Amo-Adjei, Ph.D

Academic Editor

PLOS ONE

Additional Editor Comments (optional):

Reviewers' comments:

Reviewer's Responses to Questions

**Comments to the Author**

1. If the authors have adequately addressed your comments raised in a previous round of review and you feel that this manuscript is now acceptable for publication, you may indicate that here to bypass the “Comments to the Author” section, enter your conflict of interest statement in the “Confidential to Editor” section, and submit your "Accept" recommendation.

Reviewer #1: All comments have been addressed

Reviewer #2: All comments have been addressed

2. Is the manuscript technically sound, and do the data support the conclusions?

Reviewer #1: Yes

Reviewer #2: Yes

3. Has the statistical analysis been performed appropriately and rigorously? 

Reviewer #1: Yes

Reviewer #2: N/A

4. Have the authors made all data underlying the findings in their manuscript fully available?

Reviewer #1: Yes

Reviewer #2: No

5. Is the manuscript presented in an intelligible fashion and written in standard English?

Reviewer #1: Yes

Reviewer #2: Yes

6. Review Comments to the Author

Reviewer #1: I would like to thank the authors for revising the manuscript, it is now much improved. The paper is now ready for publication.

Reviewer #2: Thanks to the authors for addressing almost all the comments I raised in the previous version. I have no further comments.

7. PLOS authors have the option to publish the peer review history of their article (what does this mean?). If published, this will include your full peer review and any attached files.

Reviewer #1: **Yes: **Gabriel O Dida

Reviewer #2: **Yes: **Abdul-Aziz Seidu

---

## [Editor Report · Acceptance letter]

21 Aug 2020

PONE-D-20-07319R1 

Barriers facing persons with disability in accessing sexual and reproductive health services in Sub-Saharan Africa: a systematic review 

Dear Dr. Ganle:

I'm pleased to inform you that your manuscript has been deemed suitable for publication in PLOS ONE. Congratulations! Your manuscript is now with our production department. 

Kind regards, 

on behalf of

Dr. Joshua Amo-Adjei 

Academic Editor

PLOS ONE